

# SIMULATING LAKE TANGANYIKA'S HYDRODYNAMICS UNDER A CHANGING CLIMATE

Kevin Sterckx[1], Philippe Delandmeter[2], Jonathan Lambrechts[3], Eric Deleersnijder[4], Wim Thiery[1,5]

[1] Vrije Universiteit Brussel, Department of Hydrology and Hydraulic Engineering, 1050 Brussels, Belgium
[2] Institute for Marine and Atmospheric Research, Utrecht University, Princetonplein 5, 3584 CC Utrecht, The Netherlands
[3] Université Catholique de Louvain, Institute of Mechanics, Materials and Civil Engineering (IMMC), 1348 Louvain-la-Neuve, Belgium
[4] Université Catholique de Louvain, Institute of Mechanics, Materials and Civil Engineering (IMMC) & Earth and Life Institute (ELI), 1348 Louvain-la-Neuve, Belgium
[5] ETH Zurich, Institute for Atmospheric and Climate Science, 8092 Zurich, Switzerland

*Correspondence to*: Kevin Sterckx (Sterckx.Kevin@outlook.com)

**Abstract.** Lake Tanganyika is the second oldest (oldest basin of the lake is 9 – 12 million years old), second deepest (1470 m) lake in the world. It holds 16% of the world's liquid freshwater. Approximately 100.000 people are directly involved in the fisheries operating from almost 800 sites along its shores. Despite the vital importance of Lake Tanganyika and other African inland waters for local communities, very little is known about the impacts of future climate change on the functioning of these lacustrine systems. This is remarkable, as projected future changes in climate and associated weather conditions are likely to influence the hydrodynamics of African water bodies, with impacts cascading into ecosystem functioning, fish availability and water quality. Here we project the future changes in the hydrodynamics of Lake Tanganyika under a high-end emission scenario using the 3D version of the Second-generation Louvain-la-Neuve Ice-ocean Model (SLIM 3D) forced by a high-resolution regional climate model. We first show the added value of 3D simulations compared to previously obtained 1D model results. The simulated interseasonal variability of the lake with this 3D model explains how the current mixing system works. A short-term present-day simulation (10 years) shows that the 75 m deep thermocline moves upward in the south of the lake until the lower layer reaches the lake surface during August and September. Two 30-year simulations have been performed (one with present day and one with future conditions), such that a comparison can be made between the current situation and the situation at the end of the 21st century. The results show that the surface water temperature increases on average by 3 ± 0.5 K. The latter influences the hydrodynamics in the top 150 m of the lake, namely the bottom of the thermocline does not longer surface. This temperature-induced stratification fully shuts down the earlier explained mixing mechanism.



# 1 Introduction

In recent decades, climate change has caused impacts on ecosystems and societies on all continents and across the oceans,

highlighting their sensitivity to a changing climate. Besides large-scale consequences, climate change will also impact the functioning of smaller-scale features, such as inland water bodies. In this paper, we will focus on the impact of climate change on the hydrodynamics of Lake Tanganyika, which is a meromictic lake (meromixis denotes the resistance to natural thermal intermixing due to the presence of a strong thermocline; see 2.4 Mixing dynamics), located between Burundi, DR Congo, Tanzania and Zambia (Fig. 1). As the countries surrounding Lake Tanganyika are mainly developing ones, people are still

strongly dependent on local sources for their basic needs, such as food, water and electricity. The lake fulfils an essential role in this supply, mainly because of the many fishery activities (Sarvala et al., 1999). There are around 100,000 people directly involved in the fisheries operating from almost 800 sites, and by estimation, 25 to 40% of the protein diet of approximately 1,000,000 people living around the lake comes from these fisheries (O'Reilly et al., 2003). Moreover, around 10,000,000 people are thought to be dependent on the lake because of its various uses (mainly fisheries and drinking water supply). The

main fishery is on pelagic clupeids (sardines), but there are many other species present in the lake and their nutrient supply is strongly dependent on the mixing of the different layers. Sardines strongly depend on the high algal productivity and an efficient carbon transfer from the algae to the fish (Tierney et al., 2010; Yvon-Durocher et al., 2012). The stratification of the water column reduces mixing between the nutrient-poor epilimnion (surface layer of a thermally stratified lake) and the nutrient-enriched hypolimnion (bottom layer of a thermally stratified lake) (De Wever et al., 2005; Paerl et al., 1975).




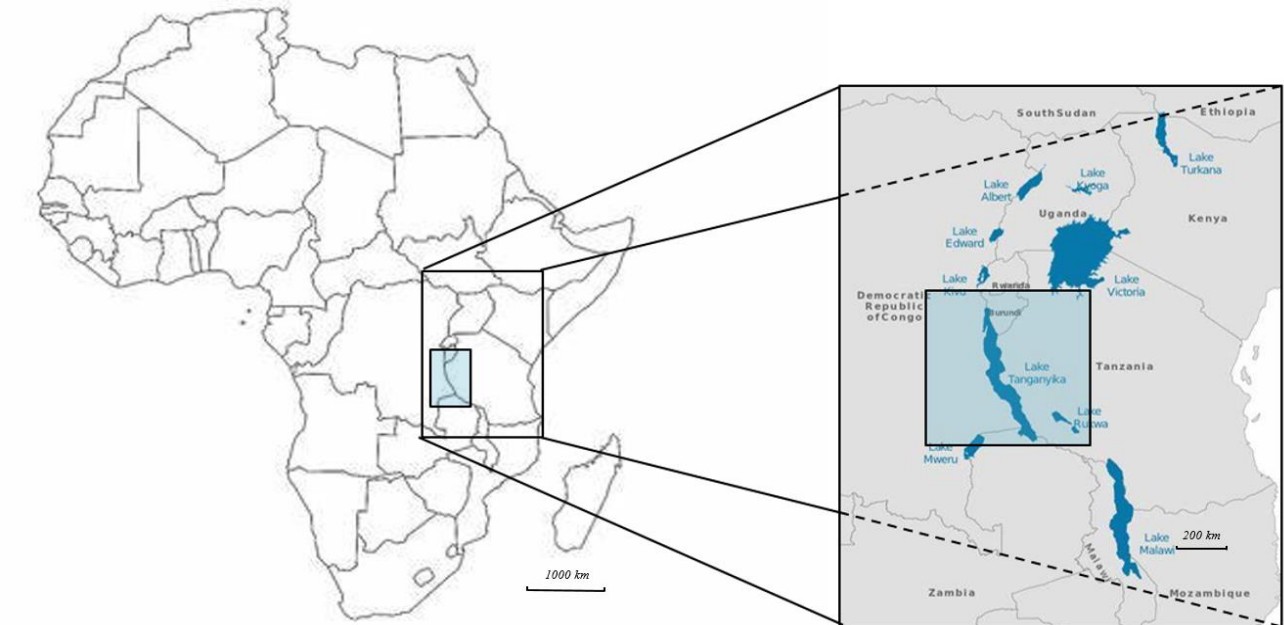

**Fig. 1: Location of Lake Tanganyika**

Past climate change has already influenced the functioning of Lake Tanganyika. This is visible through a rise water level, in surface-water temperature and an increased stability of the water column, which in turn decreases the mixing of the layers

(Kraemer et al., 2019). Historical records already suggest a 30% decrease in fish production due to climate change and its effects (O'Reilly et al., 2003). Future climate change may affect the lake system in various ways. First, atmospheric warming may potentially induce a change of the mixed layer depth and the mixing properties, which will enhance this stratification, leading to significant changes in terms of nutrient supply to the epilimnion (Kraemer et al., 2015). Stratification also affects the properties of the hypolimnion, since this layer will become more anoxic, which means the organisms that require oxygen

will be threatened. Besides atmospheric temperatures, mixing is also influenced by the wind velocity, with a decrease in wind velocity resulting in a reduction of the mixing (Naithani et al., 2011). A third influence comes from near-surface atmospheric humidity, which is inversely linked to the mixing (Wim Thiery et al., 2014; Tierney et al., 2010). A dryer atmospheric boundary layer favours evaporation from the surface water, and since this is an endothermic reaction, the water surface temperature decreases, which allows warmer water from below to rotate upwards, increasing the mixing.

However, to date very little is known about how Lake Tanganyika will respond to projected future climate change. As the atmospheric temperature is in close interaction with the lake, the atmosphere will first influence the lake temperature, which is, together with the wind driving the hydrodynamics. The extent of the feedback from the lake changes to the atmosphere is also not known. In the lake itself, the influence of hydrodynamic evolution has not been assessed regarding the chemical and biological consequences, the input parameters for the latter research can be based on the outcome of this paper.



In this study, we therefore aim to assess the present-day variability and potential future changes in the hydrodynamics of Lake Tanganyika. Specifically, this work aims to (i) assess the added value of a 3D hydrodynamic lake model compared to a 1D lake model, (ii) uncover the impact of seasonal atmospheric variability on the lake hydrodynamics by running a short-term simulation, and (iii) uncover climate change effects on the lake hydrodynamics by running two long-term simulations, in present-day and future climate conditions. To this end we use high-resolution dynamical downscalings of a reanalysis product

and a global climate model (GCM) as input for the 3D version of the Second-generation Louvain-la-Neuve Ice-ocean Model (SLIM 3D, https://www.slim-ocean.be/).

This paper is organised as follows: in Section 2 we introduce the background information regarding the Lake and the current dynamics; in Section 3, the different tools and datasets are discussed; in Section 4, the methods, including an overview with

the different runs are given; in Section 5, we show all results; in Section 6, these results are discussed; in Section 7, we formulate the final conclusions.

The acronyms used through the document are listed in Table 1:

| Acronym | Description |
|---|---|
| SLIM | Second-generation Louvain-la-Neuve Ice-ocean Model |
| CORDEX | Coordinated Regional Climate Downscaling Experiment |
| RCP | Representative Concentration Pathway |
| CMIP5 | Coupled Model Intercomparison Project |
| ITCZ | Intertropical Convergence Zone |
| ENSO | El Niño-Southern Oscillation |
| CLM | Community Land Model |
| NCAR LSM | National Center for Atmospheric Research Land Surface Model |
| PFT | Plant Functional Type |
| MPI-ESM-LR | Max-Planck-Institut für Meteorologie Earth System Model running on low resolution grid |
| GCM | General circulation model |
| ARC | Along track scanning radiometers Reprocessing for Climate |
| EVAL | Evaluation simulation |
| HIS | Historical simulation |
| FUT | Future simulation |
| IQR | Interquartile range |

**Table 1: List of acronyms (in order of appearance)**



## 2 Background

### 2.1 Lake Tanganyika

The name "Tanganyika" stands for "plain-spread lake" or "lake spreading like plain" (Stanley, 1878), and has a surface area of 32,900 km². The maximum length and width are 673 km and 72 km, respectively, while the average depth is 570 m with a maximum at 1470 m. The catchment covers an area of 231,000 km², which means it drains an area equivalent to almost the entire United Kingdom. It is the second oldest lake in the world (after Lake Baikal, Russia), where its three basins were formed over different time periods. The oldest (central) basin, began to form 9–12 million years ago, shortly followed by the northern basin (7 – 8 million years ago), and more recently the southern basin (2-4 million years ago). It is the second deepest (1470 m) lake in the world and holds the second largest anoxic volume of water (18,900 km³ or 16% of the world's freshwater) in the world after the Black Sea. The lake is located in a deep narrow trough of the western branch of the Rift Valley of East Africa between Burundi, DR Congo, Tanzania and Zambia, and it stretches from 3°20' to 8°48' S and 29°5' to 31°15' E (Fig. 1). The lake is composed of three basins, namely Kigoma (1310 m deep), Kungwe (885 m deep) and Kipili (1470 m deep) (Fig. 2), with an average width of 50 km (maximum 72 km). The shoreline has an approximate length of 1828 km at an altitude of 773 m and is mainly surrounded by steep mountains.

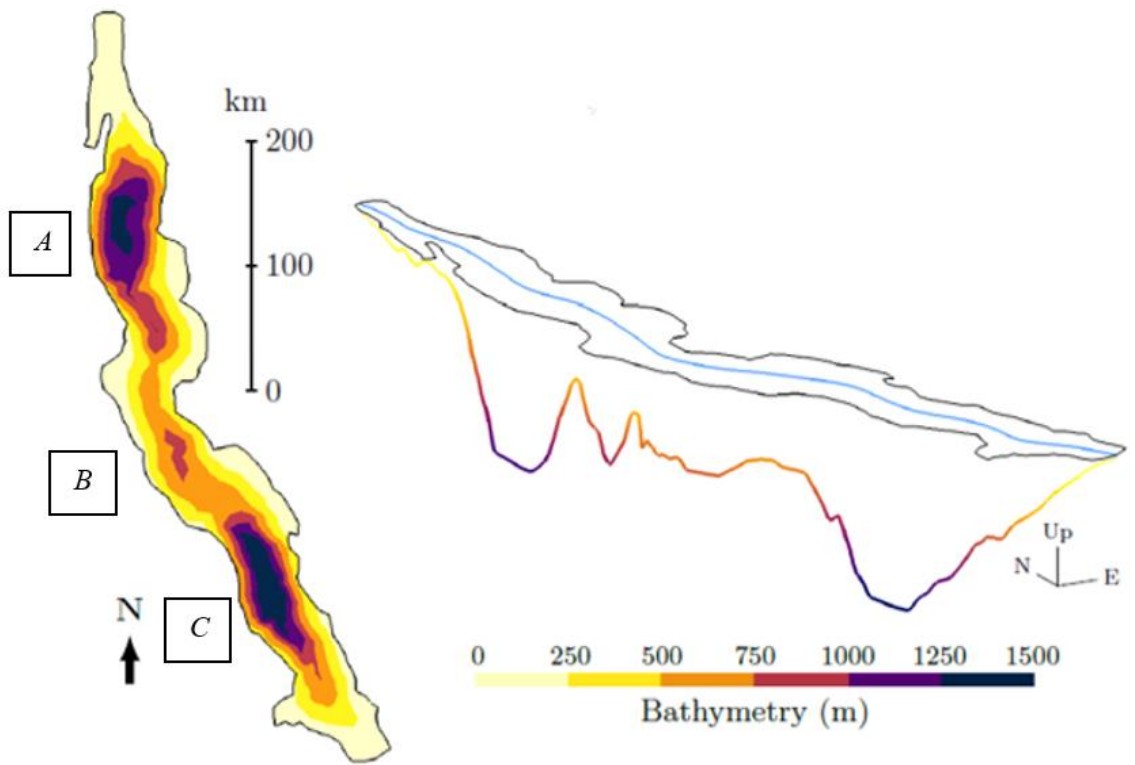


**Fig. 2: Cross section Lake Tanganyika with basins A: Kigoma, B: Kungwe, C: Kipili (Delandmeter et al., 2018)**

The location of the lake ensures – thanks to the low latitude, high temperatures and abundant light availability year-round – favourable conditions for the lake's fish productivity. The planning, management and development of the local fisheries is challenged by the fluctuations in fish catches and changes in species composition of each catch due to this unique ecosystem

(Plisnier et al., 1999).

Early biological research concluded that high water transparency, low nutrient concentrations in the epilimnion and low phytoplankton densities led to oligotrophic pelagic environment, which means the water body offers very little to sustain life (Beauchamp, 1939). Lake Tanganyika has been classified as pseudo-eutrophic (i.e. holding both eutrophic and oligotrophic characteristics) due to the local productivity of the lake by dense phytoplankton blooms and diurnal vertical movements of

zooplankton and fish. Being a eutrophic lake implies high levels of biological productivity, supported by the rich nutrient constitution (Van Meel & Kufferath, 1987). A modelling study showed that the lake is not eutrophicated (Naithani et al., 2007). In relation to climate change, air temperature measurements from around Lake Tanganyika show that the warming follows the global trend. A rise of 0.5°C to 0.7°C in air temperature has been measured between the end of the 1960's and 2019 for Lake Tanganyika, where the global evolution exhibits a rise of 0.6°C during the same period. The major increases start around the

late 1970s and coincide with the timing of regional precipitation and temperature changes (O'Reilly et al., 2003).





The Coordinated Regional Climate Downscaling Experiment (CORDEX; Giorgi et al., 2009) provides regional climate projections for Africa under various future Representative Concentration Pathways (RCPs). The results are obtained by downscaling information from Global Climate Models participating in phase 5 of the Coupled Model Intercomparison Project (CMIP5; Taylor et al., 2012). Under RCP8.5 (which is the "business as usual" scenario, where the 8.5 stands for a radiative

forcing of 8.5 W/m²), the CORDEX ensemble projects an increase in precipitation towards the coast, whereas more inland, the climate evolves to dryer conditions. Over Lake Tanganyika, which is approximately 1000 km inland, the ensemble projects a decrease in the precipitation (Souverijns et al., 2016). The latter effect might enable stronger evaporation-driven cooling, which is not included in the present study simulations.

Finally, due to climate change, a southward shift in the Intertropical Convergence Zone (ITCZ) is observed, if this were to

extend even more, the southern wind intensity in the lake's area would reduce, leading directly to reduced mixing (Tierney, 2010).

Not only climate change influences the external parameters that affect the ecosystem, also climate variability should be considered. The latter is characterized by a shorter time period (annual to multi-decadal), but has strong variations during its cycles.

**2.2 Orography and climate**

As most of the lake is surrounded by mountainous areas, the orography holds a strong role in the local climate conditions. Orographic effects include both dynamic, in which mountains disturb or distort the large-scale wind field, and thermodynamic, in which heating or cooling of mountain-slope surfaces generates flow. These orographic winds are the origin of the Föhn Effect.

Wind speed and direction over Lake Tanganyika are mainly determined by the surrounding orography. In the centre of the lake, a major speed-up occurs, which can be explained by a channelling effect along the mountain valleys. The Froude number represents the ratio of inertia forces to gravity forces, which leads to $Fr = \frac{u}{\sqrt{gL}}$, where $u = flow\ velocity, g = gravitational\ acceleration,$ and $L = characteristic\ length.$ In the south and north of the lake, the Froude number is mainly < 1, which implies orographic blocking, while the central parts of the lake allow overflow of the near-surface

(south)easterlies (Docquier et al., 2016).

In terms of meteorological behaviour, the African Great Lakes are part of one of the most complex sectors of the African continent. While the seasonal migration of the intertropical convergence zone determines precipitation seasonality, regional factors like lakes, vegetation and topography often modulate this large-scale pattern, which makes the climatic patterns and vegetation processes complex and often very local (Docquier et al., 2016; Hawinkel et al., 2016; Wim Thiery et al., 2015).

The majority of Lake Tanganyika experiences a single rainfall season, whereby the amount and timing of the precipitation may slightly vary across the lake sectors. In the northern part of the lake, intense rains occur (120 mm to 170 mm per month) from December to May. Around Tabora (central part), the intensity varies the most (60 mm to 220 mm per month). In the





southern part of Lake Tanganyika, the wet season runs from December to April with an average precipitation of 100mm to 150 mm per month, whereas from June to October the area can be considered dry (Griffiths, 1958; Savijärvi & Järvenoja,

145 2000).

At Lake Tanganyika, the long rains generally occur every year around the boreal spring (March-May), where the secondary rainfall event contains shorter rains in autumn (September-December) as those are the seasons when the ITCZ travels over the region. The interannual variability of those events are linked to atmospheric variations, including the surface temperature, monsoons, trade winds, (anti)cyclones, … The short rains are more affected by this variability and are strongly associated with

the El Niño-Southern Oscillation (ENSO) (Rasmusson & Wallace, 1983). The Southern Oscillation is the driving force behind fluctuations in the atmospheric pressure and monsoon rainfall. The warm ENSO events are usually linked to a higher amount of precipitation, where the colder ENSO events tend to result in a below average rainfall (Mutai & Ward, 2000). However, studies have shown that the ENSO impact depends on the region. For a warm ENSO, the area west of Lake Tanganyika shows poor vegetation conditions (Plisnier et al., 2000).

**2.3 Lake-atmosphere interactions**

Besides external meteorological conditions driving the lake hydrodynamics, the presence of lakes also influences the local climate in several ways. Due to the high heat capacity of water, lake rich regions experience – like coastal regions – a more moderate climate since the lakes will act as a buffer in transitioning between warmer and colder periods. The winter temperatures will experience an increase, whereas the summer temperatures will decrease. The magnitude of this effect is

depending on the isolation of the system, the size of the lakes and the initial temperature oscillations (Hostetler et al., 1994a; Huziy & Sushama, 2017). When the lake is part of a mountain system, like Lake Tanganyika, it will also significantly contribute to the precipitation in its vicinity (Gao et al., 2018; Hostetler et al., 1994b). The third meteorological parameter that is influenced by the lake is wind, influenced by orography, the presence of the ITCZ and the lake's thermal effect, where orographic winds hold the biggest share (Savijärvi, 1997). Simulations have shown that the lake's influence on the wind speed

is stronger early in the morning, and that the (daytime) lake and (night-time) land breezes, induced by presence of the lake, are enhanced by south-easterly trade winds (Wim Thiery et al., 2015; Verburg & Hecky, 2003). However, Docquier et al. (2016) also show that the influence of orography on wind speed's spatial variability has a bigger impact than the lake-land breeze system.

The final process that should be considered is the surface energy balance, as this is the driving force of the lake-atmosphere

interactions. This balance contains parameters that can be compared with the COSMO-CLM² data and will determine the stability of the atmospheric boundary layer above the lake surface and the resulting lake-atmosphere interactions. It also determines (to a large extend) the lake's heat budget by setting the upper boundary conditions, which affects the metabolism, physiology, and behaviour of aquatic organisms, since it defines the thermal structure of the lake (Saur & Anderson, 1954; Wetzel & Likens, 2000).


## 2.4 Hydrodynamics

Tanganyika is a meromictic lake with an anoxic monimolimnion, whereby meromixis denotes the resistance to natural thermal intermixing due to the presence of a strong thermocline. The thermocline dynamics mainly behave as an internal wave, governed by the topography and the atmospheric pressure and wind stress (Docquier et al., 2016; Verburg et al., 2011). During the dry season, southerly winds push the surface water northwards. Together with the increase in evapotransipration, this results in a decrease in the temperature gradient along the depth in the southern parts of the lake, where the thermocline is thus very shallow or even outcrops to the surface (Delandmeter et al., 2017). The wind events follow a periodic dynamics, with a period of 3 to 4 weeks, which are close to the first free mode of oscillation of the lake, resulting in a quasi-resonance of the internal wave (Gourgue et al., 2011; Naithani et al., 2002; Naithani & Deleersnijder, 2003).

## 3 Tools and Data

### 3.1 SLIM

The 3D component of the Second-generation Louvain-la-Neuve Ice-Ocean Model (SLIM 3D, www.slim-ocean.be) is a baroclinic model that solves the hydrostatic flow equations under the Boussinesq approximation by means of a discontinuous Galerkin finite element method on an unstructured grid (Blaise et al., 2010; Kärnä et al., 2013; White et al., 2008). It has been applied to coastal waters such as the Burdekin plume in the Great Barrier Reef (Delandmeter et al., 2015), the Columbia River region of freshwater influence (Vallaeys et al., 2018), the Congo estuary (Vallaeys et al., 2020 (submitted)) and Lake Tanganyika (Delandmeter et al., 2017). In this study, SLIM 3D is applied to Lake Tanganyika in the same setup as that of Delandmeter et al. (2017).

The most recent version of SLIM 3D runs with an estimated surface heat flux for which a relaxation term in the upper layer of the lake is use as defined in Delandmeter et al. (2017). With this relaxation term, the simulated heat will be held in the lake for a certain period, after which it is released back into the air.

Input data for this model comes from COSMO-CLM² (see 3.2 COSMO-CLM²) and will provide the required temperature and wind speed forcing.

### 3.2 COSMO-CLM²

COSMO-CLM² is a comprehensive and frequently updated regional climate model. The first building blocks of the model were a joint effort of the Consortium for Small-scale Modelling (COSMO) and the Climate Limited-area Modelling Community (clm-community), resulting in a 3D, non-hydrostatic regional climate model COSMO-CLM (Rockel et al., 2008). In COSMO-CLM² (Davin et al., 2011) the default land surface parameterisation module in COSMO-CLM, TERRA-ML, has been replaced by the Community Land Model v. 3.5 (CLM3.5; Oleson et al., 2004).



The community land model CLM is a merging between a community-developed land model focusing on biogeophysics and
an expansion of the NCAR Land Surface Model (NCAR LSM) to include the carbon cycle, vegetation dynamics, and river
routing. To represent land heterogeneity in CLM, a subgrid hierarchy has been established, where each cell gets a certain land
unit assigned, which can be glacier, wetland, vegetated, lake or urban. In the land unit sublevel, the soil properties are defined,
including colour, texture, depth and thermal conductivity. To every land unit, a second sub-grid level is assigned, called the
column, which captures the variability of the soil and snow state variables. The third sub-grid level is called the plant functional
type (PFT) and can assign up to 4 out of the 15 possible types to a certain column.

The added value of COSMO-CLM² compared to the default COSMO-CLM configuration regarding the representation of the
near-surface climate has been established for several domains including Europe (Davin et al., 2011, 2016) and Sub-Saharan
Africa (Akkermans et al., 2014; Wim Thiery et al., 2015).

Here we use three high-resolution simulations with COSMO-CLM² conducted by Thiery et al. (2015; 2016) over the African
Great Lakes region to force SLIM 3D. The first regional climate simulation, termed EVAL, consists of a reanalysis
downscaling and is available for a 13-year period (1996-2008). Results from the EVAL simulation represent the 'best guess'
of the regional climate and may be employed for model evaluation purposes and analysis of present-day spatio-temporal
patterns.

The second COSMO-CLM² simulation represents a 33-year integration (1981-2010) using information from the MPI-ESM-
LR GCM as global boundary conditions. Finally, the COSMO-CLM² simulation called FUT was obtained by downscaling
MPI-ESM-LR for the 33-year period 2068-2011 under Representative Concentration Pathway (RCP) 8.5. Comparison of the
HIS and FUT simulations enables assessment of the projected future changes in climate under a high emission scenario without
mitigation.

Each COSMO-CLM² simulation was conducted at a horizontal resolution of ~7 km (0.625°) and is nested within a COSMO-
CLM simulations run at ~50 km (0.44°) resolution in the framework of the Coordinated Regional Climate Downscaling
Experiment (CORDEX; Panitz et al., 2014). As such, COSMO-CLM² is used to downscale global-scale information from a
reanalysis or GCM to the African Great lakes region via a continental-scale intermediate nesting step. The final nesting step
has been tailored to the region by (i) applying COSMO-CLM in its tropical configuration (Panitz et al., 2014), (ii)
implementing a high horizontal resolution providing pioneering representation of the local topography and (iii) using a 1D
lake model to represent tropical lake surface water temperatures (Wim Thiery et al., 2015).

### 3.3 FLake

The 1D model that will be used to evaluate the benefits of a 3D model over a 1D model is FLake. FLake is a one-dimensional
model, which is interactively coupled to COSMO-CLM². The input of the model is similar to that of SLIM, since it demands
meteorological data, however, this time, the integral energy budget is computed by the model. To represent the water
circulation, the model considers two layers, being a surface mixed layer, and a thermocline just below. For the mixed layer,





the temperature is assumed to be uniform, where the thermocline does contain a certain temperature gradient over the depth (Thiery et al., 2015).

FLake has been proven to be comparable to other 1D lake models for different lake types and climatic conditions (Perroud et al., 2009). Besides this, there have been offline tests for several African Great Lakes, concluding that FLake shows a strong
performance for calculating lake surface temperatures (Thiery et al., 2014a,b).

## 3.4 ARC Lake

ARC stands for Along track scanning radiometers Reprocessing for Climate (Thiery et al., 2015). The radiometers are sensors which are part of the European Space Agency's Earth Observing missions. The aim of the ARC Lake is to derive observations of lake surface water temperatures and lake ice covers of major lakes for 1991 until 2010
(http://www.laketemp.net/home_ARCLake/).

## 4. Methods

When pre-processed, the COSMO-CLM² output was arranged in a well-defined order, such that it can be used as input for SLIM 3D. Three runs were performed (as shown in Table 2), namely EVAL, HIS and FUT. The evaluation simulation (EVAL) was run over a 10-year period to validate the model and investigate the effects of interseasonal meteorological variations on
the lake's hydrodynamics. The historical simulation (HIS) was run over a 30-year period, where the results were used both to evaluate the difference between the ERA-interim and GCM downscaling, and as a reference for the future simulation (FUT). The latter was also run of a 30-year period, but with a simulation start date, which is 90 years after that of HIS.

| SLIM simulation name | EVAL | HIS | FUT |
|---|---|---|---|
| Global-scale forcing for COSMO-CLM² | ERA-interim | GCM downscaling | GCM downscaling |
| Simulation period start | 01/01/1999 | 01/01/1980 | 01/01/2070 |
| Simulation period end | 31/12/2008 | 31/12/2009 | 31/12/2099 |
| Model spin-up (to be counted before the start of the simulation period) | 2 years | 12 years | 12 years |
| Meteorological spin-up (to be counted before the model spin-up) | 1 year | 1 year | 1 year |

**Table 2: Technical specifications of the three simulations performed in SLIM 3D**





## 5 Results

### 5.1 Model validation – evaluation simulation

For the first part of this work, the 3D model (SLIM 3D) has been compared with a 1D model (FLake). For both models, the
bias in surface water temperature with data from ARC Lake as reference has been calculated and plotted. In Fig. 3 (a, b, c),
the average bias on the 1D model is shown for periods of three months, grouped per season: February-March-April (FMA),
June-July-August (JJA), and October-November-December (OND). The most remarkable deviations are during the dry season
(Jun-Aug), when there is a strong underestimation of the temperature in the central and southern part of the lake. During the
first wet season (Feb-Apr), temperatures are slightly overestimated, reaching a maximum at the northern and southern ends.
Finally, during the second wet season (Oct-Dec), the warm bias shows again, but mainly in the northern regions.



**Fig. 3: Seasonal surface water temperature bias in the Flake (a-c) and EVAL (d-f) against the ARC Lake reference product (simulation – reference) for the months February-April (FMA), June-August (JJA) and October-December (OND). (g-i) Change in bias in the EVAL simulation compared to the Flake simulation.**



In Fig. 3 (d, e, f), the bias on SLIM 3D is shown, indicating that there are still some deviations compared to the ARC Lake data, but those are consistently smaller than they were for FLake. The cold bias in the south and centre during dry season are

all gone. The small warm bias in the southern tip during the first wet season became smaller and even the yearlong warm bias in the north has been reduced (although still present).

For the specific case of the centre of the lake, the 3D model considers the water velocities, and since this is a very windy area, this might explain the cold bias of the 1D model. After the validation of the 1D model, done by Thiery et al. (2015), the authors hypothesized that the system reacts too fast on seasonal meteorological changes, which means that it overshoots at every

seasonal transition. Their explanation for this was the lack of heat storage, since the 1D model underestimated the thermocline depth. In the 3D model, it is clear that there is a certain buffer, resulting in a slight delay on the heating effects, but the cooling effects are still strongly represented. It can thus be hypothesised that the breakdown of the stratification is better simulated by SLIM than its reconstruction. Another hypothesis can be the use of the relaxation function to represent the surface heat flux (Delandmeter et al., 2017), which includes a delay compared to changes in meteorological variables. In general, it can be

concluded that SLIM 3D adds value compared to FLake (1D) for simulating the lake surface water temperature, which highlights the importance of accounting for better 3D hydrodynamics Fig. 3 (g, h, i). The effect of a better 3D circulation in SLIM is clearly of higher importance than a better simulated surface heat flux in FLake. Further research should be done to assess the added value of a better heat flux representation in SLIM, since this would hypothetically further improve the quality of the model.

**5.2 Interseasonal variation – evaluation simulation**

As a next step, one can look at the annual variation of the temperatures. Fig. 4 and Fig. 5 show the water temperature over the first 150 m depth, which is the result of the EVAL simulation. The range of 150 m has been selected, as below this level, there are no seasonal effects on the temperature (Verburg & Hecky, 2009). In cross-sections like those shown in Fig. 4 and Fig. 5, some instabilities have occurred (in this case for January and November). These are strictly numerical and do not interfere

with the global understanding of the processed, nor the assessment of the model quality.

The variations are expected to be varying with latitude, which can also be seen in our results (Kraemer et al., 2015). The northern part of Lake Tanganyika is surrounded by mountains, resulting in lower wind speeds. When combining this with the high average atmospheric temperature due to its equatorial location, it explains why the highest surface water temperatures along the lake can be found here.

The southern part of the lake has deeper mixing due to the bigger contrast between the dry and wet seasons. This means that although the total average temperature of the water column is not that different, the surface water temperature will be lower.



**Fig. 4: Monthly water temperature cross-sections (EVAL) January - June**



Fig. 5: Monthly water temperature cross-sections (EVAL) July - December






The same reasoning goes for the centre of the lake, which is the most turbulent part due to the internal water transport, combined with the presence of strong winds due to the flat landscape. The combination of the latter will also result in deeper mixing, reducing the average surface water temperature. In Fig. 4 and Fig. 5, it is also shown that the middle of the lake has weaker stratification due to this.

In general, according to Naithani et al. (2002, 2003), the external temperature has little to no influence below 100m depth. This corresponds to the thermocline, never being below 75m (Fig. 4 and Fig. 5). Also, on average, the dry months have stronger lake evaporation, resulting in a higher position of the thermocline, although this is not simulated by SLIM. The same period is also colder, which reduces the surface water temperature.

### 5.3 Model validation – 30-year simulation

The second part of this work consists of longer simulations, again forced with meteorological data, but from the GCM downscaling without reanalysis. The major difference is that for the evaluation simulation, the data originates from a downscaling of the reanalysis ERA interim. This is a GCM downscaling, where every six hours, the quality has been improved by corrections on the output, based on all available measurements. Another difference is the runtime of the simulation, as it has never been tested if one or two years of spin-up are enough to obtain equilibrium in the model. Fig. 6 shows the difference

in average surface water temperature between the EVAL and HIS simulations (for their entire periods), where almost no difference can be observed.

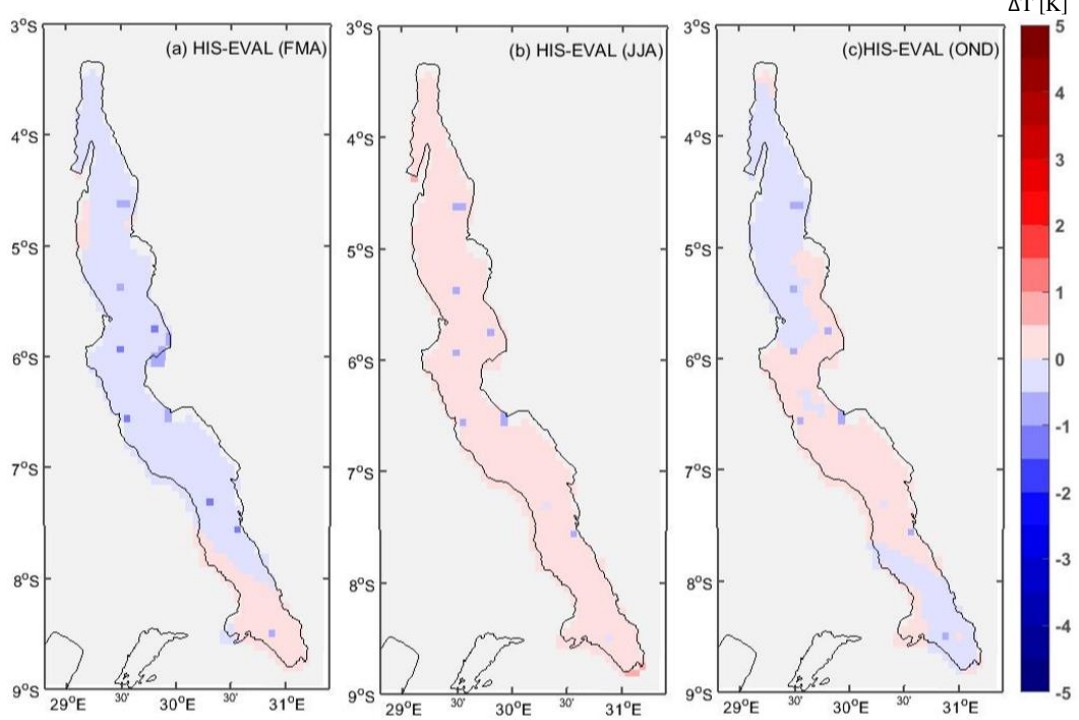

**Fig. 6: Seasonal relative surface water temperature plots.**





In Fig. 7, Fig. 8, Fig. 9 and Fig. 10, the interquartile range (IQR) of both the HIS and FUT simulations are shown, based on
both temporal and spatial variations. Fig. 7 shows the monthly mean temperatures for both simulations, where the IQR on the
spatial variation in indicated. This tells us that at any point in time, the coldest zones of the FUT simulation are still warmer
than the hottest ones of the HIS simulation. Not only the spatial variation can be represented, Fig. 8, Fig. 9 and Fig. 10 show
the temporal IQR at different locations, based on the variation of the monthly mean temperatures over the 30 years. Only in
Mpulungu, which is located at the southern tip of the lake, the IQR is +/- 1 K, which is double the value of Bujumbara and
Kigoma. However, in this area, the interseasonal temperature variation is also twice as much as in the rest of the lake, as this
is the area where the thermocline climbs up first.

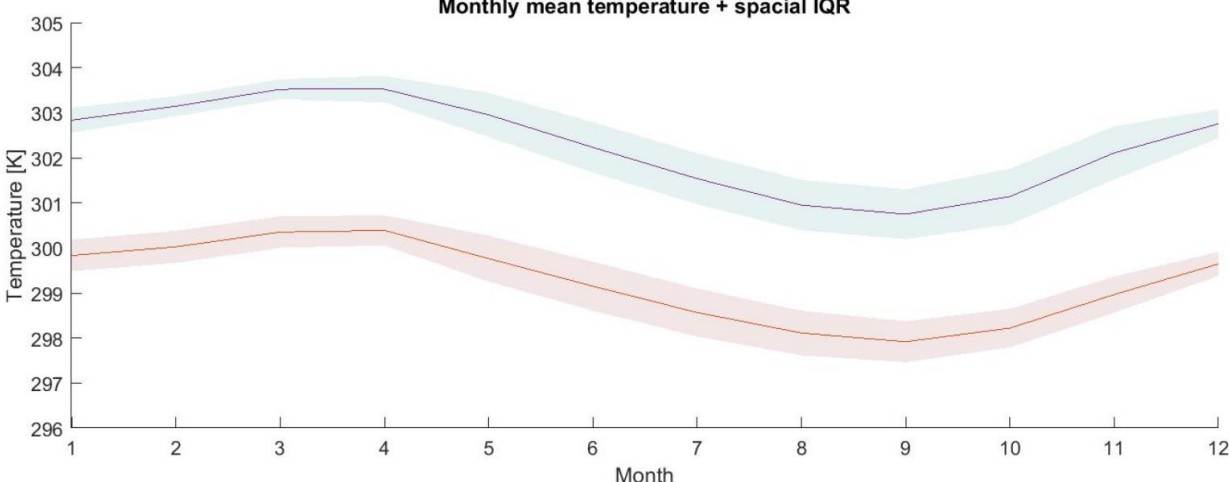

**Fig. 7: Spatial IQR on surface water temperature for HIS (red) and FUT (blue).**

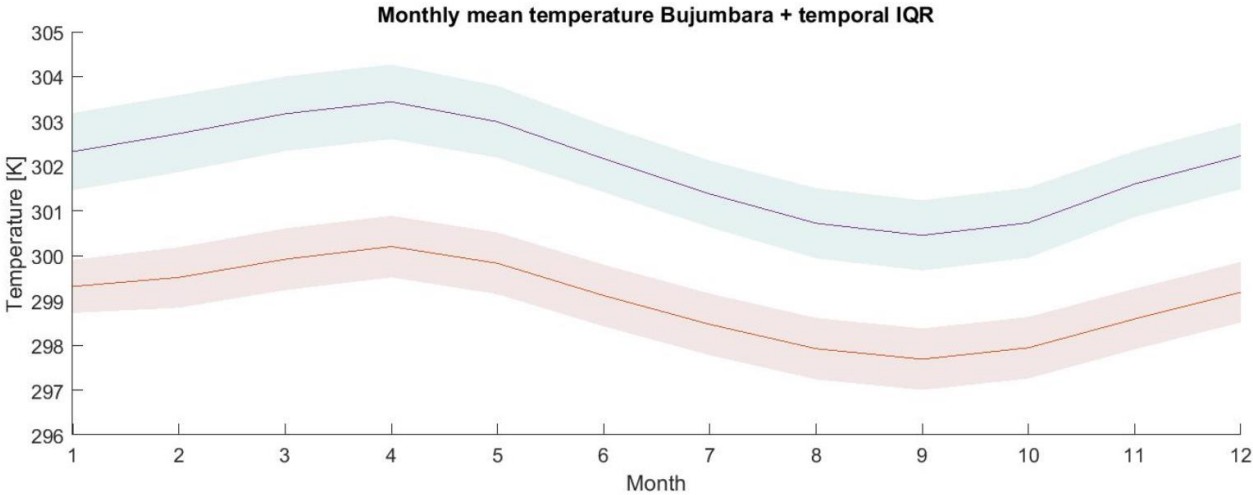

330         **Fig. 8: Temporal IQR at Bujumbara on surface water temperature for HIS (red) and FUT (blue).**





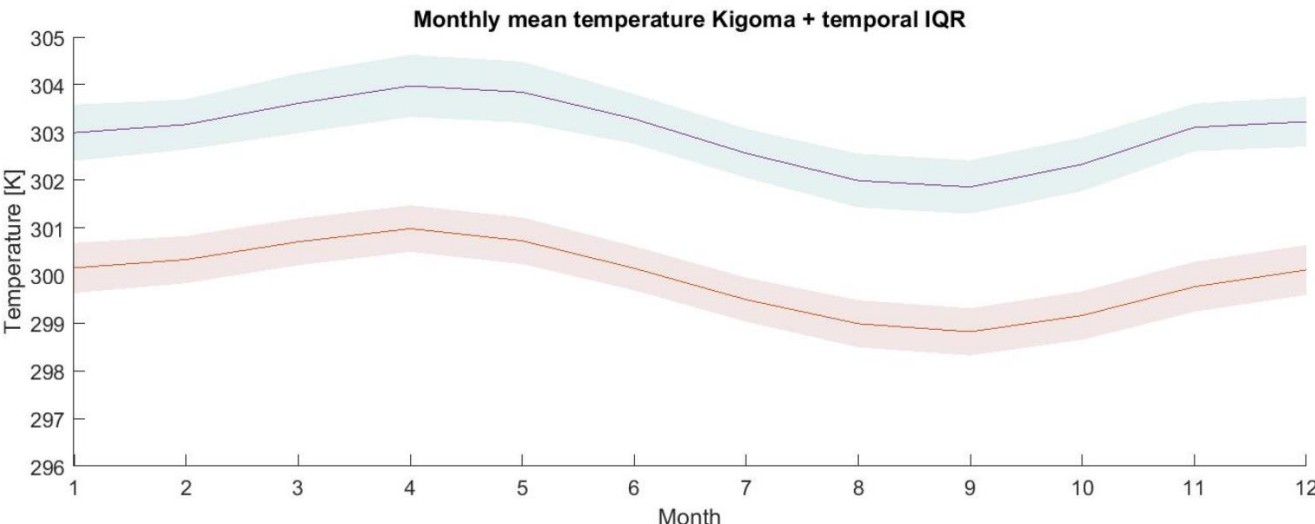

**Fig. 9: Temporal IQR at Kigoma on surface water temperature for HIS (red) and FUT (blue).**

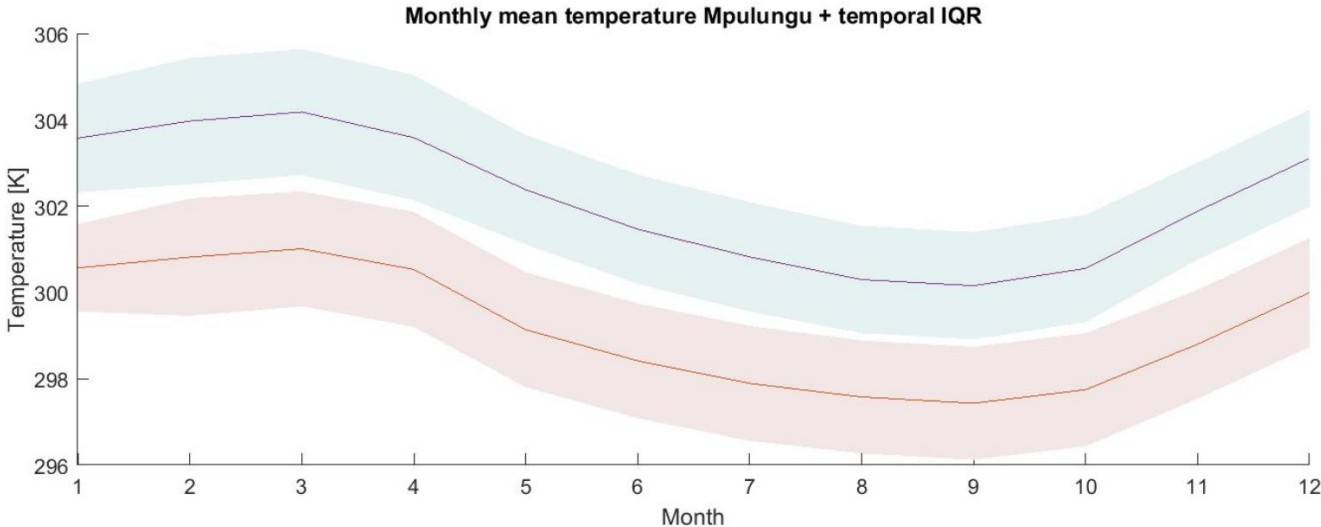

**Fig. 10: Temporal IQR at Mpulungu on surface water temperature for HIS (red) and FUT (blue).**

### 5.4 Climate change simulation – 30-year simulation

The comparison between the FUT (Fig. 12) and HIS (Fig. 11) simulations immediately shows a clear message: uniform warming (Fig. 13). The surface water temperature increases over the entire lake, over the entire year by $3 \pm 0.5$ K. A small wave of increased heat can be observed, travelling from south to north before the dry season, but since the average temperature difference is at the minimal scale, this might be within the margin of error as well.



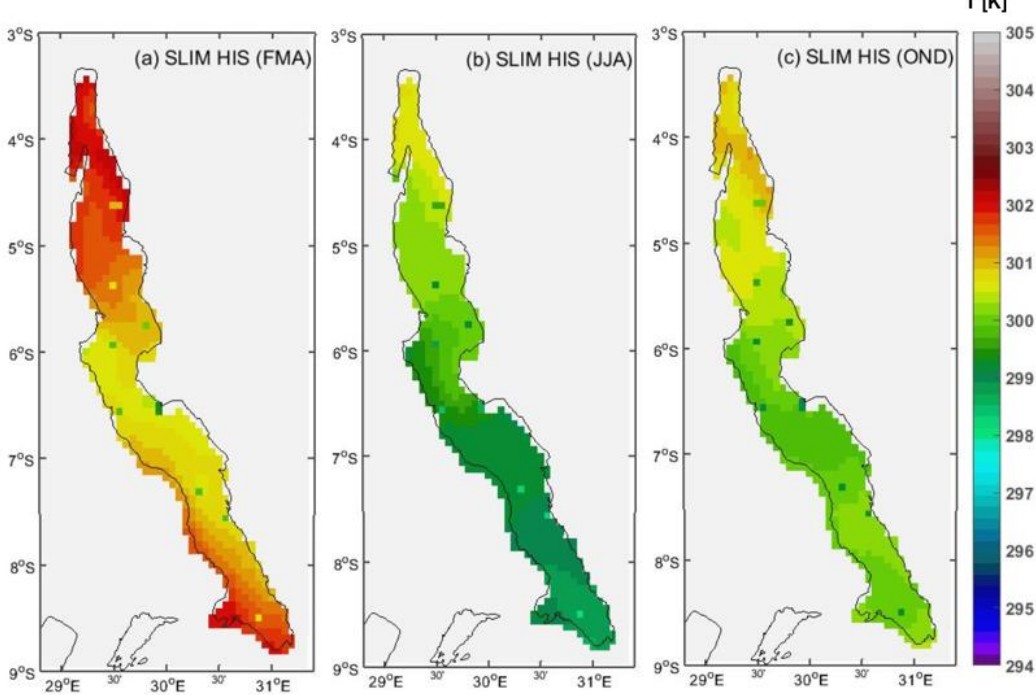

**Fig. 11: Mean seasonal surface water temperature from the HIS simulation.**

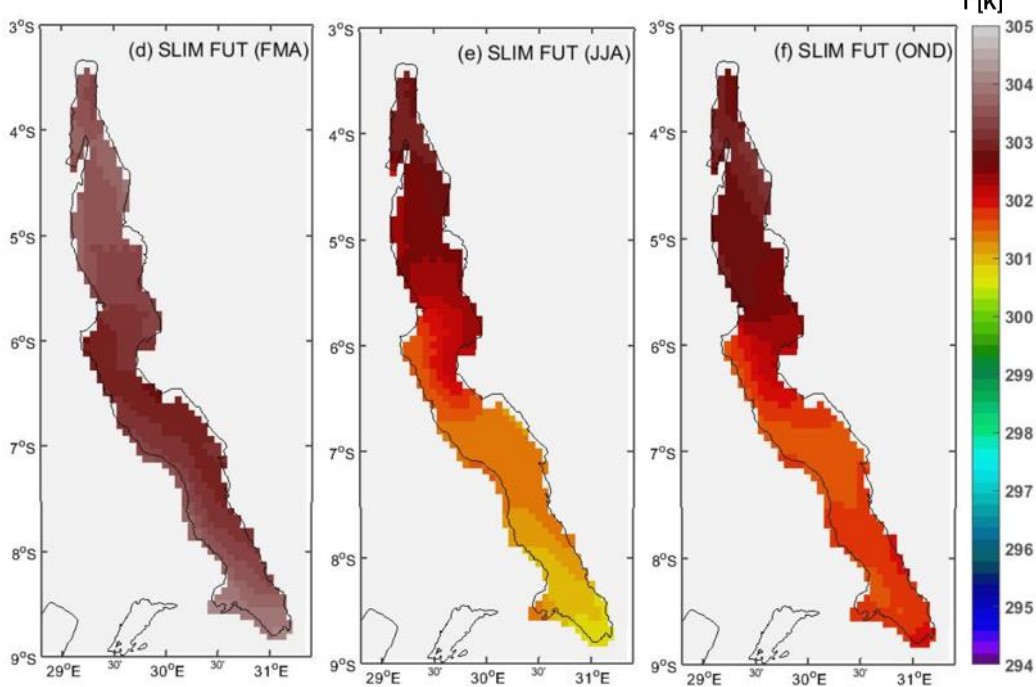

**Fig. 12: Mean seasonal surface water temperature from the FUT simulation.**


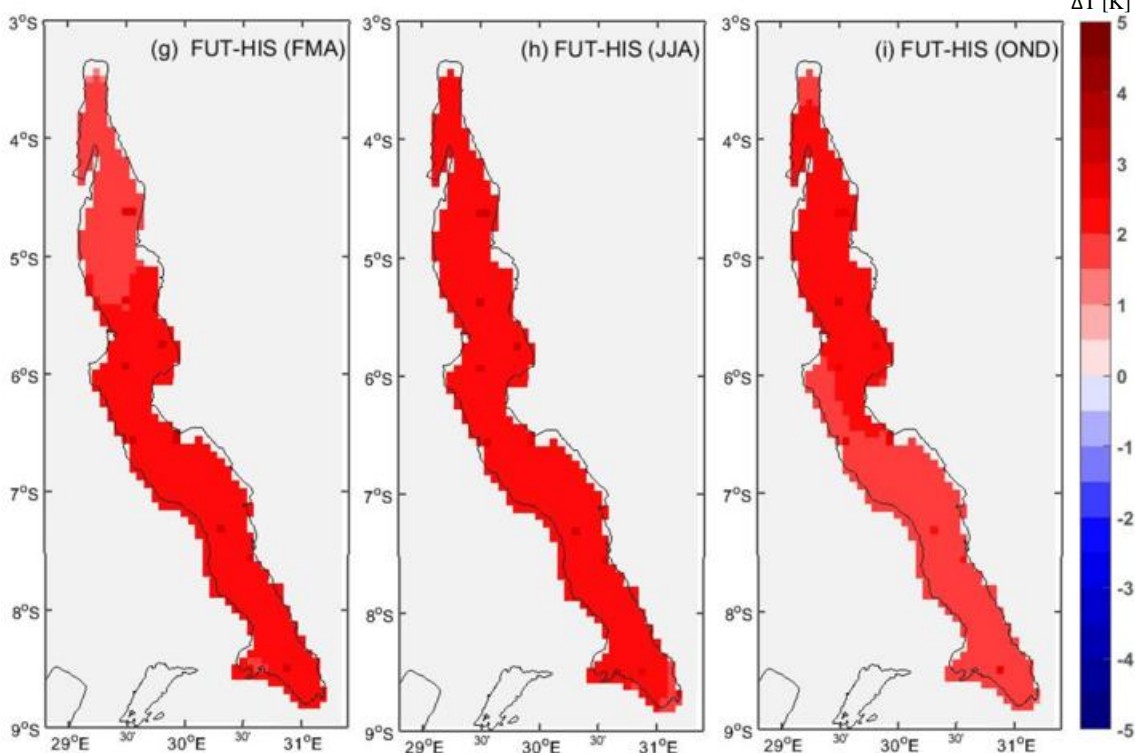

**Fig. 13: Relative seasonal surface water temperature (FUT - HIS)**

A second important observation is the conservation of the stratification during dry season, as shown in Fig. 16 and Fig. 17. Where in the current situation (Fig. 14 and Fig. 15), during dry season, the thermocline moved all the way up from the south to the centre of the lake, this appears not to be the case anymore, resulting in all year long stratification.

Finally, Fig. 18 and Fig. 19 show the difference in average temperature between the HIS and FUT simulations, plotted over the depth. It is clearly visible that the gradient overall increases, with a maximum at the surface level, resulting in a much stronger stratification of the lake. In addition, the external heating in FUT influences the water temperature approximately 75 m (dry season) to 100 m (wet season) deeper than it was the case for HIS. This effect is probably the most important of all, since it links back to the first part of the study, where the water quality and the provision of the nutrients were linked to the

strength of the stratification. This increased stratification results in a severe reduction of the mixing, and thus potentially to a strong decrease in nutrient supply to the ecosystem and eventually to the fish population.

It should be considered that the humidity is not included in this model. This might result in lower or higher water temperatures, depending on the season. If the seasonal cycle would become more pronounced, the dry season becomes dryer, resulting in more evaporation, decreasing the surface water temperature, where the wet season becomes more humid, which means the

surface water temperature can increase even more. Further research should include a better representation of the surface heat flux in SLIM, which could take this effect into account.







**Fig. 14: Monthly water temperature cross-sections (HIS) January – June**





365             **Fig. 15: Monthly water temperature cross-sections (HIS) July - December**





**Fig. 16: Monthly water temperature cross-sections (FUT) January - June**





**Fig. 17: Monthly water temperature cross-sections (FUT) July - December**




**Fig. 18: Monthly relative water temperature cross-sections (FUT - HIS) January - June**





**Fig. 19: Monthly relative water temperature cross-sections (FUT - HIS) July – December**



## 6 Discussion

Our results suggest that unabated climate change will induce important changes in the lake's mixing regime, with potentially severe implications for its ecosystem. This is because the input of nutrients in the upper layer is likely to be drastically reduced, thereby impacting phytoplankton blooms and, hence, the whole food web. However, there are papers that have shown no strong support to the hypothesised reduced phytoplankton due to warming (Kraemer et al., 2017). Clearly, the impacts of future climate change on the whole ecosystem most be further investigated.

The conclusions of this paper are based on numerical simulations, for which input data and calibration are based on other models or remote sensing. A comprehensive evaluation of the model system with in-situ measurements was already performed by Delandmeter et al. (2017), and our additional validation confirms that the model can be used for climate change projections. However, SLIM 3D still has some opportunities for improvements, such as the implementation of the surface heat flux to include the relative humidity. The input data is also obtained from one regional climate simulation, as this was linked to SLIM

3D. Other climate simulations are likely to generate different outcomes. However, the extent of these variations is not known. Moreover, this study considered only one climate change scenario (the high-emission scenario RCP 8.5), implying that we do not sample the range of potential greenhouse gas concentration scenarios that our planet may follow.

Based on the above, together with the tests, simulations and improvements that need to be done, the major effort should be to test the sensitivity of our results against a range of representative concentration pathways and GCM/RCM combinations.

## 7 Conclusions

The hydrodynamics of Lake Tanganyika is driven by the interaction between the lake and climate. External forcing from mainly wind, temperature and humidity induce variations in the water temperature and circulation patterns. The lake has been modelled before, but there has never been an in-depth analysis of the effects and consequences of climate change on the lake hydrodynamics.

The main goal of the first part of this work has been to compare SLIM 3D to the 1D model, which is interactively coupled to COSMO-CLM² (FLake) and to assess the accuracy of the output by comparing it to satellite data (ARC Lake). To this end, SLIM was forced by a reanalysis downscaling with the regional climate model COSMO-CLM² for the period 1980-1999. The results of the comparison with FLake are unambiguous and show that the overall performance of SLIM 3D is better than FLake. The comparison with ARC Lake confirms this statement but indicates that the reconstruction of the thermocline can

still be improved. From those results, it can be concluded that a 3D circulation model brings major improvements compared to a 1D water column model. Future work on this includes the implementation of a better surface heat flux in SLIM 3D, as this might improve the quality of a correct and faster reconstruction of the thermocline. However, studies have shown that the heat flux will only be properly represented, when local values of air density, kinematic air viscosity and latent heat of vaporization (Verburg & Antenucci, 2010).



The second part of the study comprises the analysis of the results of the simulation executed by SLIM 3D. This 10-year simulation shows strong reactions to seasonal activity in the lake, especially in the southern part. As the centre of the lake is not located in a montane area, winds speeds are increasing, leading to a reduction of the depth of the thermocline and improving the mixing. The north of the lake stays the hottest part, which is mainly caused by a reduced wind speed and higher average atmospheric temperatures.

For the final part of this paper, a new atmospheric forcing dataset is used to force onto SLIM 3D, which has been obtained by downscaling a global climate model (GCM) with COSMO-CLM². Before any conclusions could be drawn, the consistency of the SLIM simulation output forced by the reanalysis and GCM downscaling had to be tested. This comparison showed almost no difference, which means that moving from reanalysis to GCM forcing has a limited impact on the simulated climatology in this area.

The final projection has been done for 30 years in the future under a high-emission scenario (RCP 8.5), starting in 2070, and has been compared to a historical reference, starting in 1980. The main conclusion of this analysis is the overall warming of the surface water, both temporal (so not season dependant) and spatial (over the entire lake). It is also clear that the breakdown of the thermocline during dry season is no longer happening, which means a permanent stratification is projected for Lake Tanganyika. This conservation of the thermocline has two causes. First of all, the depth of the thermocline overall increases,
which makes it harder to move the lower layer all the way to the surface. Second, the temperature gradient in the top layer and thermocline increases, which results in not only a deeper thermocline, but also a more thermal stable one. However, the model does not include the atmospheric humidity, which could result in higher evaporation during dry seasons, leading to better mixing and lower surface water temperatures. Regardless of the latter is the increased stability of the thermocline the major hydrodynamic conclusion of this research, which concludes that, based on this model, mixing, and thereby the entire ecosystem
of the lake, might come to collapse.

## 8 Acknowledgements

The computational resources were provided by the universities that are part of the "Fédération Wallonie-Bruxelles" (Federation Wallonia-Brussels), under the consortium called CÉCI, which stands for "Consortium des Équipements de Calcul Intensif" (Consortium of intensive calculation equipment). The authors like to thank Jean-Pierre Descy for his insight on the topic.

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
