# Peer review of "SIMULATING LAKE TANGANYIKA'S HYDRODYNAMICS UNDER A CHANGING CLIMATE"

_Earth System Dynamics, 2020_

## Referee Comment (RC1) · Anonymous Referee #1 · 6 Aug 2020

GENERAL COMMENTS: This manuscript discusses the warming of Lake Tanganyika with future climate change as per the RCP8.5 scenario, as assessed through 3D numerical simulations. Though both the topic and the study case are of high relevance, this manuscript provides a too small contribution to be published on its own. As a matter of fact, despite the use of a 3D model, the study of "lake hydrodynamics", as introduced in the title, is reduced to only epilimnetic temperature variations, mostly addressing the surface only, without any detail on water circulations. Temperature variations in the hypolimnion are neglected: while they are increasingly slow with increasing depth, they nevertheless do occur. The study should have dealt with the growing inertia of deep waters through proper long-term simulations of thermal structure evolution, especially given the exceptional depth of Lake Tanganyika. Furthermore, while SLIM 3D works

great for assessing present day hydrodynamics as regards both the stratification dynamics and circulations, I have several concerns as regards its application to climate change studies, due to its incomplete modelling of lake surface heat budget. The comparison of SLIM 3D with the 1D model FLake is not correctly developed, as explained in the specific comments below. The manuscript also implicitly relies too much on the Delandmeter et al. (2017) paper: no details about the model are given herein, which puzzles the general reader. The most relevant ones, e.g. mesh resolution, should at least be provided. Methods are not clearly explained and I have some concerns over the results, some model blemishes being overly evident. The Introduction and the very small Discussion sections lack an actual insight and a comparison against the wide literature available on lake warming with climate change and the related modelling efforts. The organisation of the paper is poor and makes me think that it is an incomplete rearrangement of a M.Sc. thesis. Several language errors are also present. Because of these general reasons, in addition to the specific comments below, I must suggest the rejection of the manuscript by the journal. I hope my comments would stimulate the Authors to expand and review their work and try to publish it again soon.

SPECIFIC COMMENTS: L. 54: What is the meaning of the expression "more anoxic"? It does not make sense. L59. The expression "rotate upwards" should be replaced with something more technical. L60-65. This whole passage is unclear. L73-76. If a paper follows a standard structure, there is no need to present a recap of its content. L101-106. This whole passage is contradictory. Which is the trophic state of the lake? What is the anthropogenic influence on it? It is not clear. L130-131. Wind accelerates also because the drag coefficient of water is much lower than that of land cover. L131-135. This discussion on the Froude number of wind is unclear. L146-148. This sentence is in contradiction with those above, in which it is stated that the Lake Tanganyika area experiences a single rainfall season. L154. What is the meaning of "poor vegetation conditions"? L169-174. This whole passage is unclear. L193-195. This passage is unclear. Which part of the heat budget at the surface simulated by the model? Which part is neglected? What is the relaxation term? What is the "certain period" for which heat

is retained into the lake and then released? L232-240. Given what is stated in the paper, SLIM 3D and FLake are not proper models to hold a comparison between 1D and 3D lake models, as they neglect different relevant phenomena: SLIM 3D does not consider the full heat budget at the water surface interface, while FLake does not consider temperature variations within the epilimnion. L241-245. I do not understand this whole section. ARC Lake is never mentioned before in the paper and its use is not clearly explained here. L246-254. Given that climate change is an ongoing process and that the thermal structure of deep meromictic lakes presents a thermal inertia in the scale of years up to centuries, why didn't you perform a single simulation of the 1980-2099 time span? That way, you would have been able to properly simulate the response of Lake Tanganyika to climate evolution, including temperature dynamics within the hypolimnion. Simulating separate time spans may be acceptable for lakes which display annual mixing, but it is strongly elusive for a meromictic basin. L260-261. A 1D model is supposed to give a horizontally averaged estimation of water temperature. How does the FLake simulation compare against the horizontally averaged observed surface water temperature from ARC Lake? L268-271. This comparison does not make sense, as the two models produce different outputs and the 1D one cannot reproduce horizontal surface temperature variations by definition. The two models should be compared over a common basis, i.e. over horizontally averaged temperature. L272-276. This whole passage is unclear. L279-281. This does not make sense. See the comments above. L281-282. I would say that for a lake of the size of Lake Tanganyika, the possibility of introducing spatially heterogeneous boundary conditions in the 3D model plays a larger part than 3D circulations on reproducing spatial heterogeneity, especially if only surface temperature is accounted. Moreover, which boundary conditions were used to drive the 1D model? Were atmospheric boundary conditions averaged over the lake surface or were data at the lake deepest point considered? L288-290. What kind of instabilities are the authors referring to? It is not clear in the sentence. Figs. 4 and 5. Such large numerical instabilities (as in Fig. 4a) cannot be presented in a journal paper. The Authors should identify the causes and address them. I see that the bathy-
metrical profile of the shores is roughly sketched through large steps. Does this affect calculations? By the way, what is the resolution of the numerical mesh of the SLIM 3D model? It is not specified in the paper. Wasn't a better bathymetry available? Most importantly, where are the cross sections placed? L301-302. Why should the centre of the lake be the location of maximum turbulence? It is not clear. L310-313. Such information should be conveyed within the Methods, not in the Results. L313-314. A basic investigation should be performed to understand such issue. L314-316. This sentence is puzzling. Do the Authors imply that climate change has negligible effect on lake surface temperatures? Fig. 6. What is the reason behind the sparse blue points? I suspect there is an error of some sort in data processing or in model results. L320-321. How is the spatial IQR calculated? Which data are used to determine the distribution of results? At which time? Fig. 11. What is the reason behind the sparse green points? See the comment above. L352-353. This sentence is not clear. Figs. 14 and 16 Why do numerical instabilities always arise in January? See the comment above. L382. Which kind of additional validation was performed in this work which confirms that the model can be used for climate change projections? L399-400. How is this possible? ARC Lake provided observations of surface temperature only. L402-404. This sentence misses a verb.

---

## Referee Comment (RC2) · Anonymous Referee #2 · 10 Aug 2020

Review of the paper entitled "Simulating lake Tanganyika's hydrodynamics under a changing climate" by

The authors have investigated how the temperatures in Lake Tanganyika will evolve in response to climate change. The evolution of lakes thermal structure under climate change has been largely investigated over the last decades with countless number of publications. Yet, African lakes remain poorly investigated. Lake Tanganyika is especially interesting given his very specific features. However, I do not think this paper reaches the scientific standard to be pulished at this stage. There is no presentation of the model, no calibration and validation of the model (this is an issue regarding reproducibility). Figures are barely discussed. I counted for instance 3 lines of text for 4 figures. I encourage the authors to rework the manuscript in order to

strengthen the message. Specific comments are provided in the annotated manuscript.

Please also note the supplement to this comment:
https://esd.copernicus.org/preprints/esd-2020-36/esd-2020-36-RC2-supplement.pdf

———————————————————

[Figure]

**Supplement:**

[revised manuscript text omitted]

---

## Author Comment (AC1) · 25 Oct 2020

**Author's response to reviewers**

Kevin Sterckx1, Philippe Delandmeter2, Jonathan Lambrechts3, Eric Deleersnijder4, Piet Verburg5, Wim Thiery1,6

1 Vrije Universiteit Brussel, Department of Hydrology and Hydraulic Engineering, 1050 Brussels, Belgium

2 Institute for Marine and Atmospheric Research, Utrecht University, Princetonplein 5, 3584 CC Utrecht, The Netherlands

3Université Catholique de Louvain, Institute of Mechanics, Materials and Civil Engineering (IMMC), 1348 Louvain-la-Neuve, Belgium

4 Université Catholique de Louvain, Institute of Mechanics, Materials and Civil Engineering (IMMC) & Earth and Life Institute (ELI), 1348 Louvain-la-Neuve, Belgium

5 National Institute of Water and Atmospheric Research, 3216 Hamilton, New Zealand

6 ETH Zurich, Institute for Atmospheric and Climate Science, 8092 Zurich, Switzerland

Correspondence to: Kevin Sterckx (Sterckx.Kevin@outlook.com)

The authors would like to thank referee #1 and #2 for the time devoted to review the manuscript and for their useful and constructive suggestions. All comments by the referees were carefully addressed and the manuscript has substantially benefited from the proposed changes. Here below, we would like to clarify our changes regarding all comments, except the small comments on word usage, spacing,... as suggested by referee #2, as these were all justified and have been incorporated in the revised manuscript.

The following convention is applied in this response letter to denote modification in the original manuscript: *response to the referee*; <del>deleted words</del>; **added words**.

**Anonymous Referee #1**

GENERAL COMMENTS: This manuscript discusses the warming of Lake Tanganyika with future climate change as per the RCP8.5 scenario, as assessed through 3D numerical simulations. Though both the topic and the study case are of high relevance, this manuscript provides a too small contribution to be published on its own. As a matter of fact, despite the use of a 3D model, the study of "lake hydrodynamics", as introduced in the title, is reduced to only epilimnetic temperature variations, mostly addressing the surface only, without any detail on water circulations. Temperature variations in the hypolimnion are neglected: while they are increasingly slow with increasing depth, they nevertheless do occur. The study should have dealt with the growing inertia of deep waters through proper long-term simulations of thermal structure evolution, especially given the exceptional depth of Lake Tanganyika. Furthermore, while SLIM 3D works great for assessing present day hydrodynamics as regards both the stratification dynamics and circulations, I have several concerns as regards its application to climate change studies, due to its incomplete modelling of lake surface heat budget. The comparison of SLIM 3D with the 1D model FLake is not correctly developed, as explained in the specific comments below. The manuscript also implicitly relies too much on the

Delandmeter et al. (2017) paper: no details about the model are given herein, which puzzles the general reader. The most relevant ones, e.g. mesh resolution, should at least be provided. Methods are not clearly explained and I have some concerns over the results, some model blemishes being overly evident. The Introduction and the very small Discussion sections lack an actual insight and a comparison against the wide literature available on lake warming with climate change and the related modelling efforts. The organisation of the paper is poor and makes me think that it is an incomplete rearrangement of a M.Sc. thesis. Several language errors are also present. Because of these general reasons, in addition to the specific comments below, I must suggest the rejection of the manuscript by the journal. I hope my comments would stimulate the Authors to expand and review their work and try to publish it again soon.

Thank you for this feedback. In this response letter we address each of the comments raised. Based on the reviewer suggestions, we have made substantial changes to the initial manuscript. In short, these changes include: (i) a substantial extension of the introduction and discussion sections including reference to key recent studies on lake warming, (ii) a careful revision of the spelling in the manuscript, (iii) an update and rearrangement of most of the figures, (iv) a more detailed description of the SLIM 3D model including the surface heat flux representation and an argumentation for its applicability to future projections, (v) a clarification of the FLake simulation set-up and motivation for the comparison of both simulation tools, and (vi) the discovery of the model blemishes as a postprocessing artefact which we now removed. More details on each of these changes are provided in the step-by-step replies. To expand the discussion and interpretation of the results, Piet Verburg, an expert on Lake Tanganyika, joined the author team.

We would also like to note that, as the reviewer points out, the topic of this study is of high relevance. Our study is the first to investigate projected climate change impacts on the hydrothermal characteristics of Lake Tanganyika. Our results show dramatic projected changes in the temperature and mixing regime of this lake, which were hitherto not known. In light of the anticipated impacts of these drastic changes on ecosystem functioning and local communities, we believe that this information represents a relevant contribution to the peer-reviewed literature, despite the possible improvements to the study design and analysis which can be taken into account in follow-up studies.

**We would like to start by showing the revised introduction:**

In recent decades, climate change has caused impacts on ecosystems and societies on all continents and in the global ocean, highlighting their sensitivity to a changing climate. Climate change also affects the functioning of inland water bodies. In this study, we focus on the impact of climate change on the hydrodynamics of Lake Tanganyika, a meromictic lake located between Burundi, DR Congo, Tanzania and Zambia (Fig. 1a). With the countries surrounding Lake Tanganyika being relatively poor, their inhabitants are strongly dependent on local resources for their basic needs, such as food and water. The lake fulfils an essential role in this supply, mainly because of the intensive fishery (Sarvala et al., 1999). There are around 100,000 people directly involved in the fisheries, operating from almost 800 sites, and by estimation, 25 to 40% of the protein diet of approximately 1,000,000 people living around the lake comes from these fisheries (O'Reilly et al., 2003). The main fishery is on pelagic clupeids (sardines), and together with many other fish species in the lake, their food supply is strongly dependent on the vertical mixing between deeper and shallower water layers. Clupeids strongly depend on algal productivity and an efficient carbon transfer from the algae to fish (Tierney et al., 2010; Yvon-Durocher et al., 2012). However, stratification of the water column reduces mixing between the nutrient-poor epilimnion and the nutrient-enriched hypolimnion (De Wever et al., 2005; Paerl et al., 1975), limiting primary productivity (Verburg et al. 2003).

**Fig. 1: (a) Location of Lake Tanganyika (3°20'S - 8°48'S; 29°12'E - 31°12'E), surrounded by Burundi, Tanzania, Zambia and DR Congo; (b) Cross section Lake Tanganyika with basins A: Kigoma, B: Kungwe, C: Kipili (Delandmeter et al., 2018)**

Past climate change has already influenced the functioning of Lake Tanganyika. This is visible through a rise in water level, in surface water temperature and in stability of the water column, which in turn decreases the mixing of the layers (Kraemer et al., 2019). Future climate change is further affecting the lake system in various ways. First, atmospheric warming induces a change of the mixed layer depth and the mixing properties, which enhances stratification, leading to significant changes in terms of nutrient supply to the epilimnion (Verburg et al. 2003; Kraemer et al., 2015; Woolway and Merchant, 2019; Woolway et al., 2020; Maberly et al., 2020). Primary productivity has decreased as indicated by increased water transparency, reduced uptake of dissolved silica by diatoms and decreased phytoplankton biomass (Verburg et al., 2003). Historical records suggest a 30% decrease in fish production due to climate change and its effects (O'Reilly et al., 2003). Also, reduced vertical mixing may result in a shallower oxycline with anoxic water closer to the surface which could result in more frequent fish kills (Coulter, 1963) by nearshore upwelling of anoxic hypolimnetic water. Upwelling may occur less frequent in the future but may have greater impact when it does happen (Lau et al. 2020). Besides, by atmospheric temperature, mixing is also influenced by wind velocity, with decreasing wind velocity resulting in reduced mixing (Naithani et al., 2011). A third influence comes from near-surface atmospheric humidity, which is inversely linked to the mixing (Verbur and Antenucci 2010; Thiery et al., 2014). A dryer atmospheric boundary layer enhances evaporation from the surface water, and since this is an endothermic reaction, the water surface temperature decreases, which drives the natural convection. This means that the different temperatures in the water cause density variations, leading to buoyancy forces. In this case, it will lead to upwelling of the underlying warmer water, driving the mixing mechanism. Effects on the lake surface temperature are translated into impacts on hydrodynamics.

However, to date very little is known about how Lake Tanganyika will respond to projected future climate change. The extent of the feedback from future changes in the lake to changes in the atmosphere is also not known. A key parameter influencing the lake is atmospheric temperature, as it directly influences the lake surface water temperature. A second parameter is wind speed, as this acts as a catalyser for evaporation and upwelling. Third, changes in atmospheric humidity may affect the evaporation potential and thereby the surface temperature. Finally, changes in shortwave and longwave radiation may further affect the surface energy budget. The outcomes of this paper represent a starting point for further research on projected ecological changes.

In this study, we therefore aim to assess the present-day variability and potential future changes in the hydrodynamics of Lake Tanganyika. Specifically, this work aims to (i) assess the added value of a 3D hydrodynamic lake model compared to a 1D lake model, (ii) uncover the impact of present-day seasonal atmospheric variability on the lake hydrodynamics by analysing a short-term simulation, and (iii) uncover climate change effects on the lake hydrodynamics by analysing two long-term simulations representing in present-day and future climate conditions. To this end, we use high-resolution dynamical downscalings of a reanalysis product and a global climate model (GCM) as input for the 3D version of the Second-generation Louvain-la-Neuve Ice-ocean Model (SLIM 3D, https://www.slim-ocean.be/).

**L. 54. What is the meaning of the expression "more anoxic"? It does not make sense.**

Thank you for your comment. We corrected the manuscript as follows:

Stratification also affects the properties of the hypolimnion, since this layer will become more anoxic, which means the organisms that require oxygen will be threatened. Also, reduced vertical mixing may result in a shallower oxycline with anoxic water closer to the surface which could result in more frequent fish kills (Coulter, 1963) by nearshore upwelling of anoxic hypolimnetic water. Upwelling may occur less frequent in the future but may have greater impact when it does happen (Lau et al., 2020).

**L59.** The expression "rotate upwards" should be replaced with something more technical.**

Thank you for your comment. We have corrected the manuscript as follows:

A dryer atmospheric boundary layer favours evaporation from the surface water, and since this is an endothermic reaction, the water surface temperature decreases, which <del>allows</del> <del>warmer water from below to rotate upwards, increasing the mixing.</del> drives the natural convection. This means that the different temperatures in the water cause density variations, leading to buoyancy forces. In this case, it will lead to upwelling of the underlying warmer water, driving the mixing mechanism.

**L60-65. This whole passage is unclear.**

Thank you for your comment. We have corrected the manuscript as follows:

However, to date very little is known about how Lake Tanganyika will respond to projected future climate change. The extent of the feedback from future changes in the lake to changes in the atmosphere is also not known. A key parameter influencing the lake is atmospheric temperature, as it directly influences the lake surface water temperature. A second parameter is wind speed, as this acts as a catalyser for evaporation and upwelling. Third, changes in atmospheric humidity may affect the evaporation potential and thereby the surface temperature. Finally, changes in shortwave and longwave radiation may further affect the surface energy budget. As the atmospheric temperature is in close interaction with the lake, the atmosphere will first influence the lake temperature, which is, together with the wind driving the hydrodynamics. The extent of the feedback from the lake changes to the atmosphere is also not known. In the lake itself, the influence of hydrodynamic evolution has not been assessed, regarding the chemical and biological consequences, the input parameters for the latter research can be based on the outcomes of this paper. The outcomes of this paper represent a starting point for further research on projected ecological changes.

L73-76. If a paper follows a standard structure, there is no need to present a recap of its content.

Thank you for this suggestion. We have removed the paragraph:

This paper is organised as follows: in Section 2 we introduce the background information regarding the Lake and the current dynamics; in Section 3, the different tools and datasets are discussed; in Section 4, the methods, including an overview with the different runs are given; in Section 5, we show all results; in Section 6, these results are discussed; in Section 7, we formulate the final conclusions.

**L101-106. This whole passage is contradictory. Which is the trophic state of the lake? What is the anthropogenic influence on it? It is not clear.**

Thank you for your comment. To elaborate, according to Van Meel, L. I. J., & Kufferath, J. (1987), the lake is pseudo-eutrophic. According to the concentrations of chlorophyll a and nutrients the lake can be classified as oligotrophic. On the other hand, the lake's productivity is relatively high for such an oligotrophic lake, in part as a result of rapid recycling within the epilimnion. We have edited the paragraph as follows:

Early biological research concluded that high water transparency, low nutrient concentrations in the epilimnion and low phytoplankton densities led to suggest the lake is oligotrophic pelagic environment, which means the water body offers very little to sustain life (Beauchamp, 1939). Lake Tanganyika has been classified as pseudo-eutrophic (i.e. holding both eutrophic and oligotrophic characteristics) due to the local the relatively high productivity of the lake by dense phytoplankton blooms and diurnal vertical movements of zooplankton and fish. Being a eutrophic lake implies high levels of biological productivity, supported by the rich nutrient constitution (Van Meel & Kufferath, 1987). A modelling study showed that the lake is not eutrophicated (Naithani et al., 2007).

**L130-131. Wind accelerates also because the drag coefficient of water is much lower than that of land cover.**

Thank you for this input, we suggest to add it as follows:

In the centre of the lake, a major speed-up occurs, which can be explained by a channelling effect along the mountain valleys **and the decrease of the drag coefficient over water compared to land.**

**L131-135. This discussion on the Froude number of wind is unclear.**

Thank you for this comment, as per suggestion of referee #2, we removed the passage:

The Froude number represents the ratio of inertia forces to gravity forces, which leads to, where and In the south and north of the lake, the Froude number is mainly < 1, which implies orographic blocking, while the central parts of the lake allow overflow of the near-surface (south)easterlies (Docquier et al., 2016).

L146-148. This sentence is in contradiction with those above, in which it is stated that the Lake Tanganyika area experiences a single rainfall season.

Thank you for pointing this out, we have rephrased this sentence as follows:

At Lake Tanganyika, the beginning of the rain season is driven by the southward migration of the ITCZ, which starts around September/October and lasts until December. This period is characterized by short rains, whereas, after a short period of decreased rainfall (January - February), a longer rains occur the long rains generally occur every year around the boreal spring (March-May), which are driven by the northward migration of the ITCZ. where the secondary rainfall event contains shorter rains in autumn (September-December) as those are the seasons when These periods are the effect of the ITCZ travels over the region.

**L154. What is the meaning of "poor vegetation conditions"?**

Poor vegetation conditions refer to the limited blooming of the vegetation during warm ENSO events. Plisnier et al., 2000 states: "For example, during warm ENSO phases, the Lake Victoria area showed warmer and more humid conditions with an increased vegetation activity while the central and southern part of the study area showed warmer, but drier conditions with a decreased vegetation activity. The area west of Lake Tanganyika is characterized by poor vegetation conditions during warm ENSO events."

This is confirmed by an analysis using satellite-based NDVI data and multiple gridded precipitation data sets (Hawinkel et al., 2016). The results notably show a negative correlation between ENSO index and monthly precipitation west of Lake Tanganyika (Illustration 1), with lower precipitation leading to lower vegetation activity in the region (i.e. positive correlation between precipitation and ENSO; Illustration 2).

---

## Author Comment (AC2) · 25 Oct 2020

The authors would like to thank referee #1 and #2 for the time devoted to review the manuscript and for their useful and constructive suggestions. All comments by the referees were carefully addressed and the manuscript has substantially benefited from the proposed changes. Please find the detailed answers in the supplement.

Please also note the supplement to this comment: https://esd.copernicus.org/preprints/esd-2020-36/esd-2020-36-AC2-supplement.pdf
* * *